# Functionalizing tandem mass tags for streamlining click-based quantitative chemoproteomics
Nikolas R. Burton [1,2] & Keriann M. Backus [1,2,3,4,5,6] ✉

Mapping the ligandability or potential druggability of all proteins in the human proteome is a central goal of mass spectrometry-based covalent chemoproteomics. Achieving this ambitious objective requires high throughput and high coverage sample preparation and liquid chromatography-tandem mass spectrometry analysis for hundreds to thousands of reactive compounds and chemical probes. Conducting chemoproteomic screens at this scale benefits from technical innovations that achieve increased sample throughput. Here we realize this vision by establishing the silane-based cleavable linkers for isotopically-labeled proteomics-tandem mass tag (sCIP-TMT) proteomic platform, which is distinguished by early sample pooling that increases sample preparation throughput. sCIP-TMT pairs a custom click-compatible sCIP capture reagent that is readily functionalized in high yield with commercially available TMT reagents. Synthesis and benchmarking of a 10-plex set of sCIP-TMT reveal a substantial decrease in sample preparation time together with high coverage and high accuracy quantification. By screening a focused set of four cysteine-reactive electrophiles, we demonstrate the utility of sCIP-TMT for chemoproteomic target hunting, identifying 789 total liganded cysteines. Distinguished by its compatibility with established enrichment and quantification protocols, we expect sCIP-TMT will readily translate to a wide range of covalent chemoproteomic applications.

Mass spectrometry-based quantitative chemoproteomics is an enabling technology for functional biology and drug discovery. Showcasing the widespread impact of chemoproteomics, recent studies have uncovered covalent degraders[1–6], novel targets with anti-bacterial activity[7,8], pinpointed redox-sensitive cysteines[9–14], mapped small-molecule-protein binding sites[15–22], and discovered latent electrophiles[23,24]. A key objective of established chemoproteomics platforms is the proteome-wide identification of the protein targets and specific residues modified by covalent chemical probes, which can serve as the launch point for drug development campaigns. Towards this objective, many research groups focus on technical innovations in three key areas: (1) covalent labeling chemistries, (2) improved sample preparation workflows that improve coverage and reduce sample loss, and (3) decreased instrument acquisition time through improved instrumentation and sample multiplexing.

Substantial advances have been made in the development of covalent labeling chemistries. Chemoproteomics platforms are now available that analyze reversible binders[21,22,25] and map all nucleophilic amino-acid side chains[26], including serine[27–29], lysine[17,30–32], tyrosine[31,33,34], methionine[35,36], aspartate and glutamate[37–39], arginine[40], and cysteine[9,16,41,42]. While these exciting advances in chemical probe technology have improved our understanding of the landscape of ligandable or potentially druggable proteomes, cysteine residues remain favored sites for drug development efforts. This favoritism is driven by the cysteine's numerous functional activities[43], the availability of proven cysteine-modifying chemistries, and the established clinical efficacy of FDA-approved drugs[44–47].

Alongside this considerable progress in covalent labeling chemistries, substantial inroads have been made into improved sample preparation and data analysis workflows. Exemplifying these improvements, our recent studies have demonstrated the utility of single-pot, solid-phase enhanced sample preparation (SP3)[48,49] for achieving increased coverage using low proteome inputs[50,51]. Innovative software such as pLink[52,53], MSFragger[54,55], and SAGE[56] have substantially decreased data processing time. Automated

[1]Department of Biological Chemistry, David Geffen School of Medicine, UCLA, Los Angeles CA, USA. [2]Department of Chemistry and Biochemistry, UCLA, Los Angeles, CA, USA. [3]Molecular Biology Institute, UCLA, Los Angeles, CA, USA. [4]DOE Institute for Genomics and Proteomics, UCLA, Los Angeles, CA, USA. [5]Eli and Edythe Broad Center of Regenerative Medicine and Stem Cell Research, UCLA, Los Angeles, CA, USA. [6]Jonsson Comprehensive Cancer Center, UCLA, Los Angeles, CA, USA. ✉e-mail: kbackus@mednet.ucla.edu

processing workflows now allow for rapid preparation of samples in 96- and 384-well plate format[57–60]. Such substantially increased capacity to rapidly prepare large numbers of chemoproteomic samples, which is essential for screening larger compound libraries, demands equal improvements in sample acquisition speed.

Together with advances in acquisition afforded by new instrumentation[61,62], isobaric labeling is a commonly employed strategy for decreasing acquisition time. Isobaric labels, such as the commercially available isobaric tags for relative and absolute quantitation (iTRAQ)[63], tandem mass tags (TMT)[64,65], and custom reagents, such as dimethylleucine[66–68] reagents, allow multiplexing of up to 21 samples at once. Most isobaric reagents feature amine-reactive groups, such as NHS-ester or triazine-ester, which incorporate the mass balancer and reporter by reacting with peptides N termini and lysine side chains. Amine-reactive mass tags have significantly enhanced data acquisition speeds with methods such as streamlined cysteine activity-based protein profiling (SLC-ABPP)[15]. Additionally, these tags have shown widespread utility for chemoproteomic applications, including uncovering ligand–protein interactions with thermal proteome profiling[69], screening of large compound libraries[15], discovering novel disease biomarkers[70–72], and uncovering differential protein expression in COVID-19 patients[73]. Hyperplexing with isotopically differentiated desthiobiotin reagents has achieved an impressive 36-plex sample throughput[74]. These studies all rely on the same general workflow: (1) cysteine biotinylation, (2) tryptic digest, (3) enrichment and isobaric labeling, and (4) liquid chromatography-tandem mass spectrometry (LC-MS/MS) analysis. The comparatively late isobaric labeling step, which occurs after sequence-specific proteolysis, is an unavoidable feature of these workflows, which introduces increased sample-sample variance and prolongs sample processing time (Fig. 1A).

As illustrated by our own silane-based cleavable isotopically labeled proteomics (sCIP) method[75] and the recently reported azidoTMT method[76], an alternative strategy is to introduce the isobaric label earlier in sample preparation via a fully functionalized "clickable" handle that features the built-in capacity for sample enrichment. The key advance of the sCIP platform was our fully functionalized enrichment reagents that contain biotin, a chemically cleavable dialkoxydiphenylsilane (DADPS) group, an azide for copper-catalyzed azide-alkyne cycloaddition (CuAAC or "click") enrichment, and an isobaric label. Thus, sCIP allowed for the incorporation of the isobaric label prior to trypsin digest, comparatively early in the sample preparation workflow. However, a limitation of the sCIP approach was its comparatively small 6-plex multiplexing. Addressing this limitation, the recently reported azidoTMT platform achieved 11-plex multiplexing with anti-TMT antibody-based peptide enrichment. Furthermore, the azidoTMT platform demonstrated improved coverage and decreased coefficient of variance when compared to the prior peptide-based isobaric labeling strategy. While highly enabling, the absence of antibody-based reagents for TMTPro together with reports of variable performance of the anti-TMT resin[76,77], highlight the still unmet need for robust and easily implementable enrichment-based isobaric labeling reagents.

Enabled by the solid-phase compatible DADPS-Fmoc reagent that was pivotal for the synthesis of our aforementioned sCIP reagents, here we establish the sCIP-TMT platform. The sCIP-TMT platform utilizes a minimalist sCIP reagent that can be in situ functionalized by TMT to achieve streptavidin-based cysteine chemoproteomics. In sCIP-TMT, the TMT reagents are conjugated to alkyne-labeled proteins via click chemistry, which allows for early sample pooling prior to proteolytic digestion (Fig. 1B). Demonstrating the utility of sCIP-TMT, here we employed a TMT10plex™-based platform for cysteine-reactive electrophilic fragment screening, which identified >19,000 cysteines on >5900 proteins across all sCIP-TMT10plex datasets. The decreased sample preparation time, compatibility with established sample preparation workflows and analysis pipelines, and anticipated compatibility with a wide range of chemical probes and scalability beyond 10-plex distinguish the sCIP-TMT platform from prior approaches.

## Results
### Synthesis of sCIP-TMT reagents
To enable our envisioned sCIP-TMT platform, we focused first on the synthesis of customized free N termini containing sCIP reagent. Guided by the amine-based labeling strategy used to generate the Azido-TMT

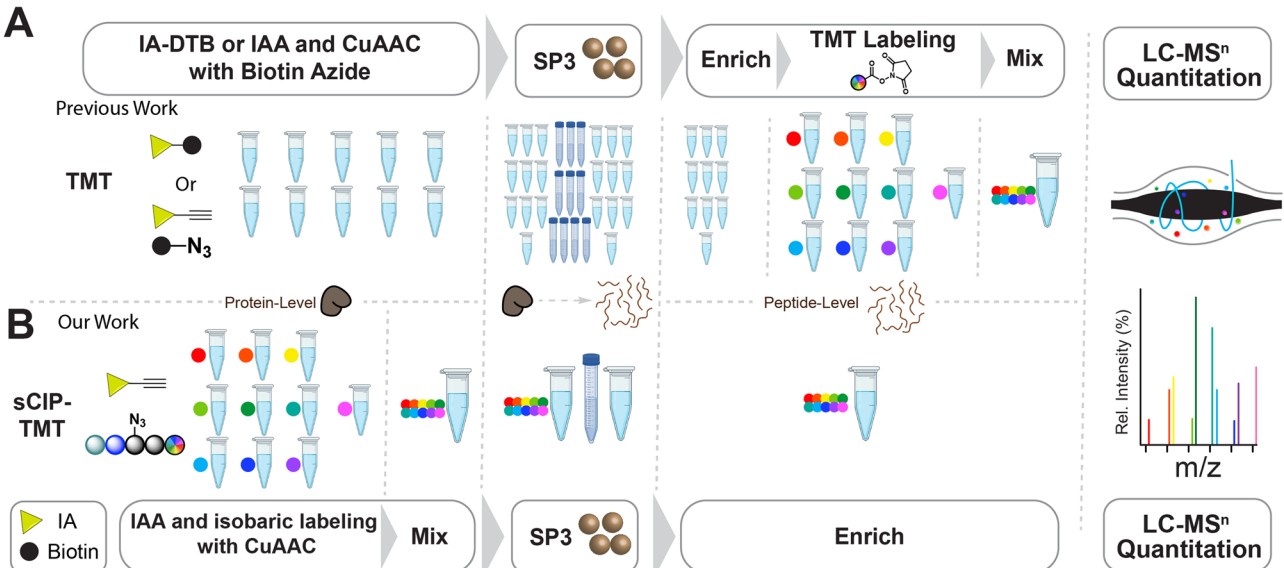

**Fig. 1 | sCIP-TMT allows for more efficient sample preparation with less sample-to-sample variance. A** Workflow currently used for profiling cysteines in which samples are labeled with either iodoacetamide-desthiobiotin (IA-DTB) or iodoacetamide alkyne (IAA) and conjugated to biotin azide via copper-catalyzed azide-alkyne cycloaddition (CuAAC or 'click'). After sample cleanup, using single-pot solid-phase enhanced sample-preparation (SP3), as illustrated here, or other decontamination methodologies, the samples are then subjected to sequence-specific proteolytic digest, isobaric labeling, avidin enrichment sample pooling, and liquid chromatography-tandem mass spectrometry (LC-MS/MS) analysis. **B** Our envisioned sCIP-TMT workflow in which fully functionalized isobaric, biotin- and azide-containing reagents allow for early-stage sample pooling directly after click conjugation. Subsequently, the labeled samples can be processed and analyzed following the established sample preparation workflow.

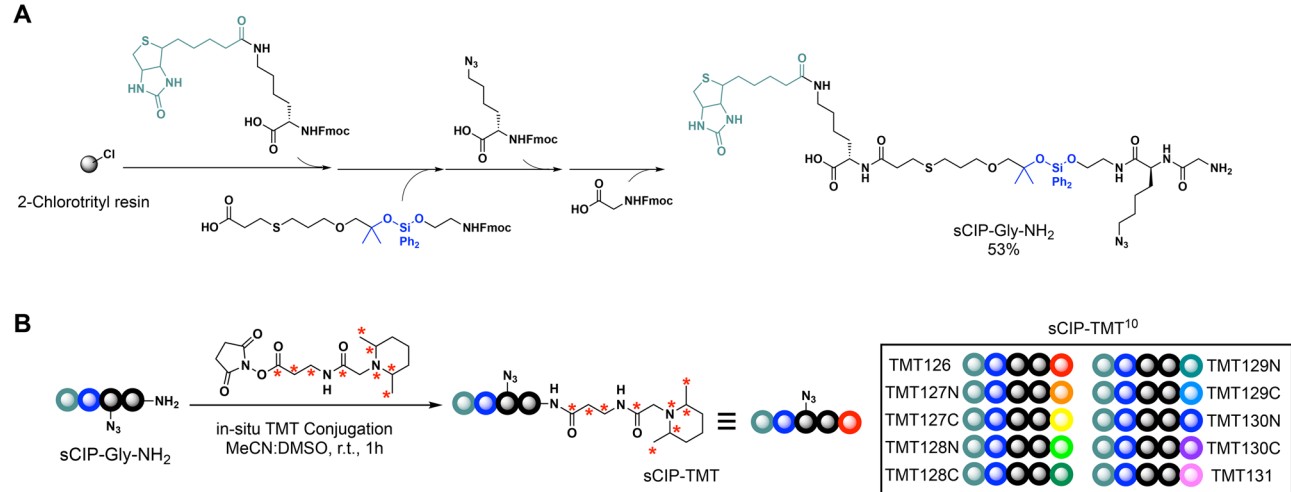

**Fig. 2 | sCIP-TMT is readily prepared in situ. A** Solid-phase peptide synthesis enables the formation of sCIP capture reagent with free N terminus (sCIP–Gly–NH₂) in 53% yield that can be **B** used to form sCIP-TMT reagents in situ by mixing sCIP–Gly–NH₂ with TMT reagents in a 1:1 ratio at ambient temperature. This method was applied to form the sCIP-TMT conjugates with the commercially available TMT10plex™ isobaric tags.

reagents[76], we envisioned that such a reagent could be easily subjected to late-stage functionalization with commercially available activated ester reagents. Enabled by our previously described solid-phase compatible DADPS building block[75], solid-phase peptide synthesis (SPPS) proceeded smoothly, yielding the final capture reagent (sCIP–Gly–NH₂) in 53% yield and high purity (Figs. 2A and S1). Of note, glycine was included as a spacer to minimize steric hindrance and to facilitate high-yield conjugation with costly isobaric reagents.

Reagent in hand, we next assessed the formation of the sCIP-isobaric conjugate by liquid chromatography-mass spectrometry (LC-MS). We selected TMT for our first-generation reagents—this isobaric reagent selection was guided by the widespread use of TMT together with reagent cost. Gratifyingly, we observed >99% conversion to the desired sCIP-TMT reagent for the reaction between TMTzero and sCIP–Gly–NH₂, with reagents mixed at 1:1 stoichiometry (Figs. 2B and S2). Notably, we opted to include a short incubation with 0.5 equivalents of hydroxylamine after sCIP-TMT conjugation to quench excess TMT reagent, following established precedent[78].

### sCIP-TMTzero achieves high-coverage cysteine labeling

Having demonstrated the highly efficient formation of sCIP-TMTzero, we next assessed reagent performance in chemoproteomics. We selected our established cysteine profiling workflow for benchmarking[11,50,51,75,79–82]. Following the workflow shown in Fig. 1B, cell lysates were capped with the pan-cysteine reactive iodoacetamide alkyne (IAA) probe (500 μM, 1 h) followed by click conjugation to the preformed sCIP-TMTzero conjugate. After sequence-specific proteolysis, enrichment, DADPS-cleavage, and peptide elution (Fig. 3A), LC-MS/MS analysis identified 3856 total proteins, 11,219 total peptides, and 8543 total unique cysteines (Fig. 3B). Aggregate analysis of sCIP-TMTzero modified peptides revealed overall higher charge states when compared to unmodified peptides, likely stemming from the added mass of the modification (+633.3957 Da) and the addition of the protonatable piperidine portion of the TMT modification (Fig. S3).

As our prior studies had revealed the formation of fragment ions derived from chemoproteomics-modified peptides[75,79], we additionally opted to perform diagnostic ion mining analysis[83] on our sCIP-TMTzero-labeled sample. We found the TMTzero reporter (m/z 126.1277) was the dominant ion identified in nearly 100% of modified spectra, having an average intensity >80% (Supplementary Data 1). Interestingly, this analysis also identified an additional diagnostic ion with m/z of 668.3896 that was frequently detected in modified PSMs (>97% of total PSMs), with a moderate 70% mean intensity. We attributed this ion to the desulfurization of

labeled cysteines (Fig. S4), which parallels the recent report of such desulfurization for peptides labeled with electrophilic compounds[84]. Inspection of the mass spectra using FragPipe-PDV[85,86] confirmed the presence of these fragment ions with the TMT reporter being the dominant ion in nearly all spectra (>91%) (Fig. 3C).

Collision energy ramping revealed maximum relative reporter ion intensity, together with maximum peptide, cysteine, peptide, and protein coverage using higher energy c-trap dissociation and normalized collision energy (NCE) of 35% (Figs. 3B and 2D). As a 36% NCE is widely reported as optimal for MS2-based TMT experiments[58,87,88], these findings support that the sCIP functionality does not substantially change the behavior of the piperidine reporter ion. Interestingly, the cysteine desulfurization ion is predominant at lower NCEs (Fig. 3E), and the TMT reporter ion predominates above 30% NCE. The high occurrence and intensity of the TMT reporter combined with the lack of other major fragments support the preferential release of the TMT fragment ion when compared to fragmentation at other points in the sCIP modification.

### sCIP-TMT10 reagents achieve high coverage and accurate quantification

Motivated by the high coverage and favorable reporter ion fragmentation observed for the sCIP-TMTzero reagent datasets, we next extended our method to TMT10Plex™. The ten sCIP-TMT conjugates were pre-formed (full reagent structures in Fig. S5) and cysteine functionalization of cell lysates was performed using IAA and click chemistry for all ten sCIP-TMT reagents in parallel. Immediately following the click reaction, the samples were pooled and subjected to cysteine chemoproteomic sample preparation, following the workflow shown in Fig. 1B. Consistent with our sCIP-TMTzero analysis (Fig. 3B), high overall proteomic coverage was achieved (Fig. 4A) for samples mixed at equimolar reagent concentrations (e.g., 1:1). This coverage, which was obtained using a 3 h gradient for acquisition, is comparable to that reported for similar studies that analyzed TMT-labeling of cysteine peptides in bulk without extensive offline fractionation[15,89].

Highlighting the streamlined nature of the sCIP-TMT workflow, we anticipate a >6 h reduction in sample preparation time together with decreased container usage (86 less sample containers) when compared to established 18-plex TMTpro workflows[15,19,42] (Fig. 4B and Supplementary Data 2). As TMT labeling prior to sample enrichment is a common strategy[15,19,42,90] as is the use of automated liquid handling[91,92], we do acknowledge that similar time savings can be achieved using these alternative and complementary strategies. Notably, our method uses comparable amounts of TMT reagent for cost-efficient TMT labeling[93].

**Fig. 3 | Defining the acquisition parameters for sCIP-TMT. A** sCIP-TMT sample preparation workflow. Cysteines are first capped using the pan-reactive molecule iodoacetamide alkyne (IAA) and then clicked to sCIP-TMTzero (pre-formed from sCIP and TMTzero as described in Fig. 2). Samples are then subjected to single-pot solid-phase enhanced sample preparation (SP3), enzymatic digestion, streptavidin enrichment, and then cleaved off resin at the DADPS moiety with acid. Upon higher energy c-trap dissociation (HCD) fragmentation the TMT reporter ion can be observed in MS/MS spectra. **B** Peptide, unique cysteine, and protein coverage of sCIP-TMTzero-labeled samples at varying HCD normalized collision energies (NCEs). **C** Representative spectra from sCIP-TMTzero-labeled peptide visualized with FragPipe proteomics data viewer (FragPipe-PDV)[85,86] showing the TMT reporter as the dominant ion present. Relative intensity of the **D** TMT reporter ion and **E** cysteine desulfurization ion at varying HCD NCEs. For panels **B–E**, $n = 1$ biological replicate per collision energy tested. All MS data can be found in Supplementary Data 1.

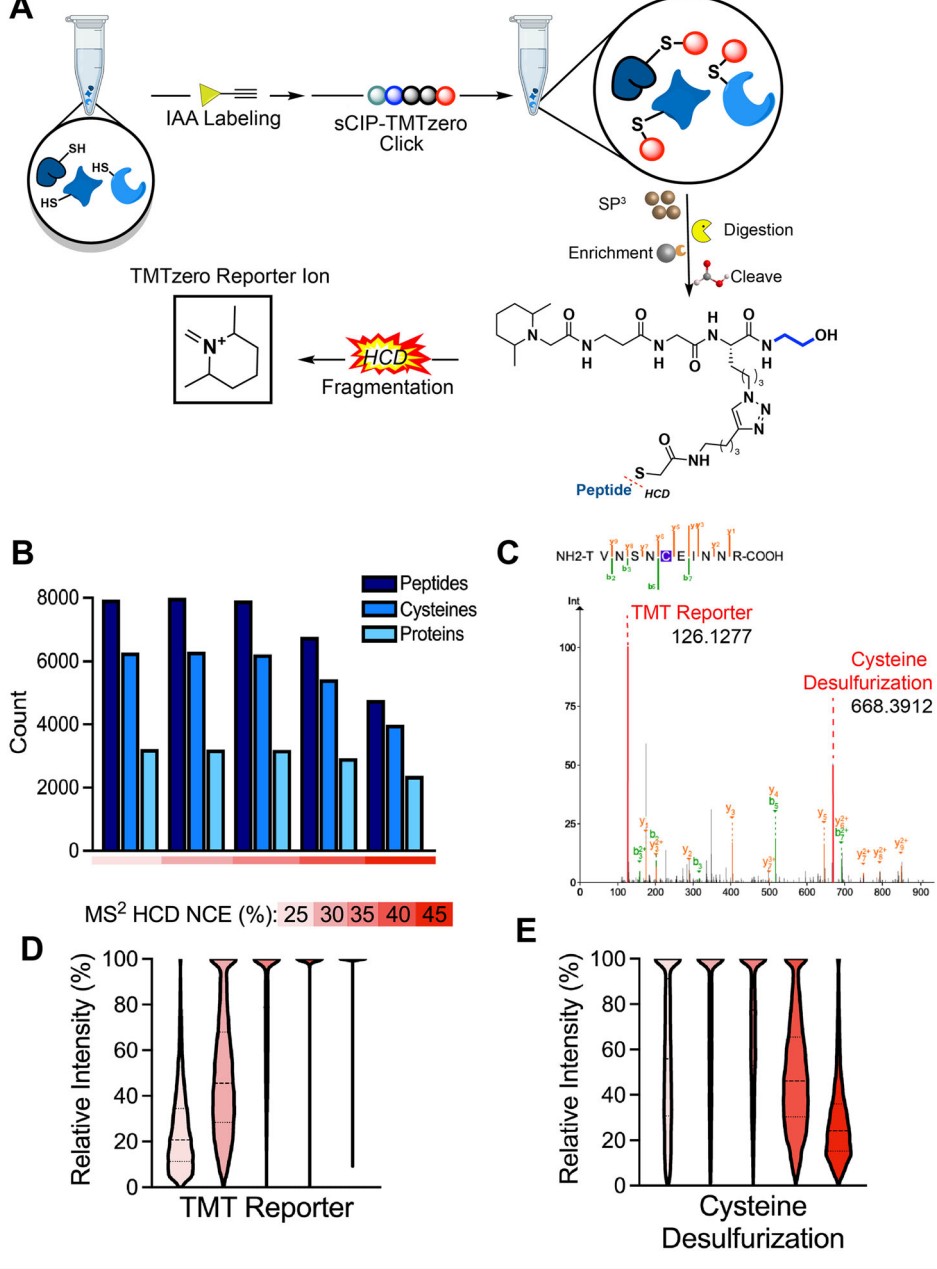

Guided by prior benchmarking of isobaric reagent performance[68,75,94], we also opted to assess the fidelity of the sCIP-TMT platform in measuring relative cysteine peptide abundance. Cysteine chemoproteomic studies are generally performed in a competitive format in which cysteine labeling sites are inferred from blockade of IAA labeling, thus we were particularly interested in vetting sCIP-TMT's capacity to quantify comparatively large fold changes. Therefore, we subjected peptides labeled with each sCIP-TMT[10] to spike-in analysis using the sample ratios indicated in Fig. 4C, D and Table S1. After LC-MS/MS analysis with a high-field asymmetric waveform spectrometry (FAIMS) device[95], the sCIP-TMT spectral files were analyzed with MSFragger software using the preset TMT workflow freely available in the FragPipe GUI[54,55,96]. We observe generally high coverage of modified peptides for all labeled samples, comparable to that obtained by the established SLC-ABPP method[15]. Importantly, the measured reporter ion intensity ratios were observed to closely match the expected values (Fig. 4C). The intensity ratios centered around one for all 10 reporters mixed in equal ratios, and the expected ratios were additionally observed for samples mixed in 1:5:10:15 proportions.

While FAIMS acquisition has proven useful for achieving a balance between high coverage and decreased ratio compression[97,98], MS3-based analysis with synchronous precursor selection (SPS)[99,100] remains the gold standard to isobaric analysis. Therefore, we additionally subjected our spike-in samples to SPS-MS3 analysis. Consistent with prior reports[101–103], this acquisition mode afforded a tighter ratio spread (Fig. 4D) together with decreased cysteine peptide coverage (Figs. S6 and S7).

## sCIP-TMT achieves highly reproducible quantification

Motivated by the high coverage and accurate quantification observed for our sCIP-TMT reagents, together with the prior reports of decreased sample variability for the related azidoTMT platform[76], we next opted to benchmark the sCIP-TMT method against samples prepared using cysteine capping with iodoacetamide-desthiobiotin (IA-DTB) and peptide labeling with TMT[19,78], show in Fig. 1, to directly compare sCIP-TMT to samples prepared with an IA-DTB and TMT. We generated and analyzed samples containing 1:1 and 1:4 ratios of peptides, functionalized with either sCIP-TMT126 and sCIP-TMT127N or TMT126 and TMT127N, respectively. This comparison

**Fig. 4 | sCIP-TMT faithfully quantifies cysteine ratios with decreased sample preparation times.**
**A** Peptide, cysteine, and protein coverage of sCIP-TMT[10]-labeled samples mixed 1:1 and analyzed using FAIMS-MS2. **B** Analysis of time (hours) and tubes saved using the sCIP-TMT workflow as multiplex channels increase. **C** Comparison of ratios for samples mixed in both 1:1 and 1:5:10:15 ratios analyzed using FAIMS-MS2. **D** Comparison of ratios for samples mixed in both 1:1 and 1:5:10:15 ratios analyzed using SPS-MS3. **E** Comparison of sCIP-TMT and IA-DTB/TMT coefficient of variance for samples mixed in 1:1 or 1:4 ratios. **F** Percentage of enriched cysteine peptides that contain either a sCIP-TMT modification (green), a TMT modification (purple), or TMT and IA-DTB modification (striped, purple). Box plots display the 5th percentile, first quartile (Q1), median, third quartile (Q3), and 95th percentile values of the sample. Error bars on bar plots display standard deviation. For panels **C–E**, $n = 3$ biological replicates. All MS data can be found in Supplementary Data 2 and 3.

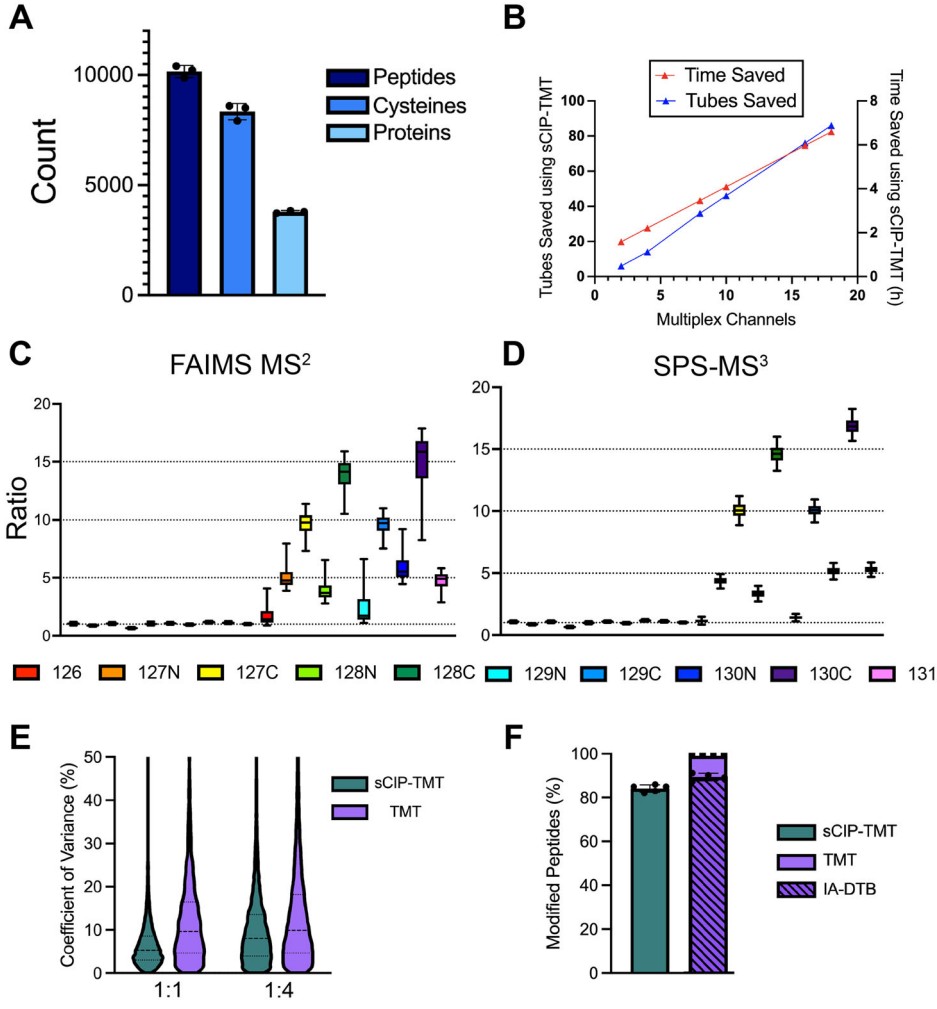

revealed ratios close to expected values for both the sCIP-TMT and TMT workflows together with increased coverage for sCIP-TMT with an equal starting protein input (80 µg) (Fig. S8A). Consistent with the reported performance of the azidoTMT reagents[76], we similarly observed a decreased ratio spread for sCIP-TMT samples (Fig. S8B). The median coefficient of variances for the 1:1 and 1:4 sCIP-TMT samples were 5.2% and 7.8%, which compares favorably to the 9.6% and 9.7% variance for the 1:1 and 1:4 TMT-prepared samples, respectively (Fig. 4E). Supporting no major performance differences between the two workflows, similar capture efficiency was observed for both workflows, with >80% of all identified peptides harboring both biotin/desthiobiotin modification (Fig. 4F). Consistent with highly efficient TMT derivatization, >99% of all peptides in the IA-DTB samples were modified with TMT, including peptides lacking the desthiobiotin modification, which are likely captured due to non-specific binding to the streptavidin resin.

## sCIP-TMT is compatible with covalent fragment screening via chemoproteomics

As cysteine chemoproteomics is widely utilized in pinpointing ligandable or potentially druggable cysteine residues, we next assess the compatibility of sCIP-TMT with screening applications. We selected four prototype electrophilic fragments (Fig. 5A), including two chloroacetamide-containing molecules, the widely utilized **KB02**[16,42,104] and **KB10**, which we had previously found showed a substantially distinct labeling pattern and more attenuated reactivity when compared to **KB02**. We additionally selected methylphenyl propiolate (**MPP**) and methyl cinnamate (**MC**), as our recent study had revealed distinct proteomic reactivity for each molecule, with

**MPP** functioning as a potent cysteine protease inhibitor whereas, **MC** showed negligible protease inhibitory activity[82].

HEK293T cell lysates were subjected to either vehicle (DMSO) or each compound (500 µM) in duplicate. Compound treatments were performed in cell lysates to avoid the recently reported pervasive protein aggregation observed in cell-based analysis using comparatively high doses of electrophilic compounds[81]. After treatment, the lysates were subjected to our sCIP-TMT workflow (Figs. 1B and S9). In total, 10733 cysteine peptides corresponding to 8515 unique cysteines, and 3787 proteins were identified (Fig. S10). 789 high-confidence cysteines were detected with log2 ratios >1 for at least one compound, consistent with covalent modification at these sites.

## sCIP-TMT reveals the proteome-wide reactivity of different cysteine-reactive electrophiles

The modest four-member compound library assayed here was selected to include a diverse set of electrophiles, which we expected to show distinct proteome-wide reactivity and target engagement profiles. To test this hypothesis, we next compared both the relative proteome-reactivity of each compound member (assessed based on the fraction of total cysteines with log2 ratios >1) and the SAR of our library members across the proteome. Quantification of the percent of total cysteines with $\log_2$ ratios >1 for each compound, which is an established proxy for overall compound reactivity[16], revealed the generally high reactivity of the chloroacetamide-containing compound **KB02**, which liganded 11.4% of total cysteines. Consistent with our prior findings[16] that the chloroacetamide-containing compound **KB10** exhibits more tempered cysteine reactivity, 3.6% of cysteines liganded by this compound in our sCIP-TMT dataset (Fig. 5B). Unlike the **MC**

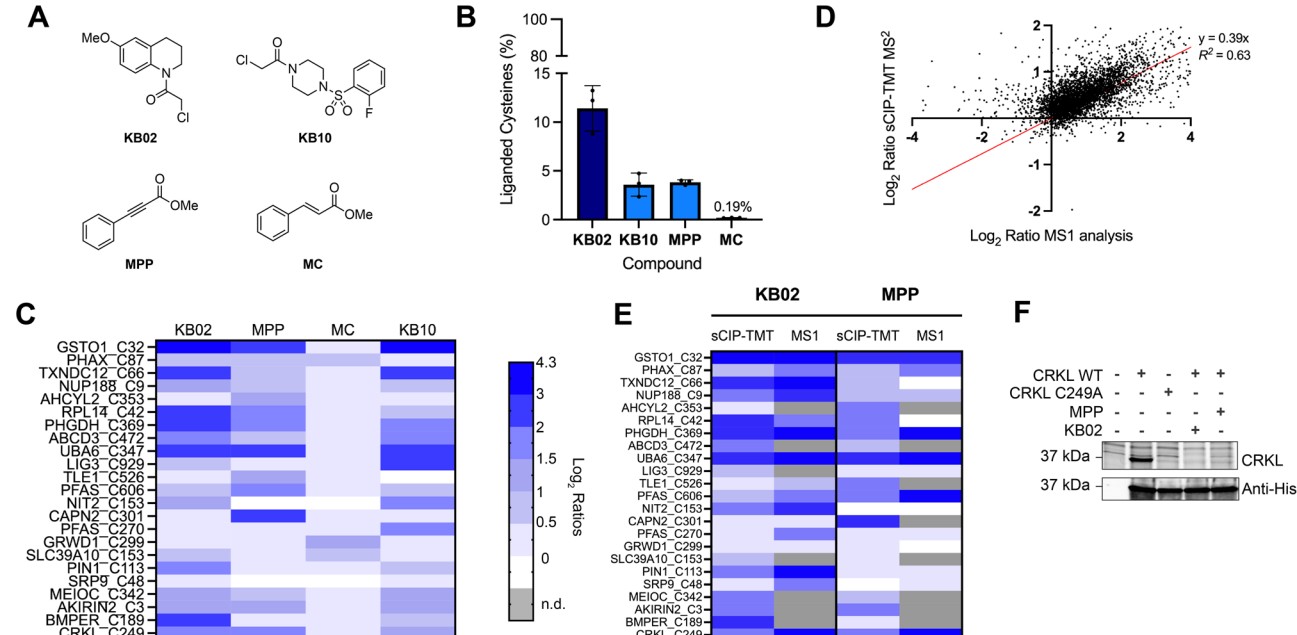

**Fig. 5 | sCIP-TMT is compatible with small-molecule electrophile screening.**
**A** Structures of electrophilic fragments analyzed by sCIP-TMT10. **B** The reactivity ratio for each compound is calculated as the number of liganded cysteines for each compound out of the total number of cysteines. **C** Heat map showing the structure–activity relationship of the four compounds across a panel of cysteines. **D** Comparison of the Log2 ratios for cysteines identified using MS1 analysis as previously reported[75] (x-axis) versus sCIP-TMT (y-axis) with scout fragment **KB02**. **E** Comparison of the ratios for cysteines shown in panel (**C**) quantified using either sCIP-TMT or an MS1-based method. For **KB02** the sCIP-TMT data was compared to our previous study using sCIP MS1 reagents[75]. For **MPP** the sCIP-TMT data was compared to our previous study using isotopic tandem orthogonal proteolysis—

activity-based protein profiling (isoTOP-ABPP)[82]. Gray boxes indicate no ratio due to no channel intensities. **F** Competitive gel-based ABPP assay to visualize KB02 and MPP cysteine reactivity with CRKL Cys249. HEK293T cell lysates were either spiked with CRKL WT or CRKL C249A point mutant (3 μM) followed by KB02 or MPP labeling (500 μM, 1 h). Samples were then incubated with iodoacetamide rhodamine (IA-Rho) (5 μM, 20 min). A decrease in-gel fluorescence for the CRKL band was observed for both the C249A mutant and the compound treated lanes indicating the cysteine is necessary for labeling by IA-Rho and **KB02** and **MPP** fully label that cysteine. For **B**, Error bars on bar plots display standard deviation. For panels **B**–**E**, n = 3 biological replicates. All MS data can be found in Supplementary Data 4.

molecule, which showed very attenuated proteome-wide reactivity (liganding 0.2% of all detected cysteines), **MPP** shows comparable cysteine reactivity to **KB10**, engaging 3.8% of all cysteines (Fig. 5B). Consistent with our prior observation that MPP engages cysteines typically labeled by chloroacetamides[82], we observe >85% of cysteines engaged by **MPP** are also engaged by **KB02** or **KB10** (Fig. S11). The capacity of **MPP** to engage cysteines labeled by chloroacetamides is further exemplified by glutathione S-transferase omega 1 (GSTO1) Cys32 (Fig. 5C), which has consistently shown strong labeling by most chloroacetamide reagents and negligible labeling by structurally matched acrylamide-containing compounds[104].

As is to be expected for a small compound screen of simple fragment electrophiles, such as ours here, we observe a high overlap between liganded targets (Fig. S11), we do observe 493 total cysteines that are uniquely modified by only a single compound. Exemplary cysteines that show strong scaffold-dependent SAR include phosphoribosylformylglycinamidine synthase (PFAS) Cys270 and DNA ligase 3 (LIG3) Cys929 uniquely labeled by **KB10** and Calpain-2 catalytic subunit (CAPN2) Cys301 and Transducin-like enhancer protein 1 (TLE1) Cys526 uniquely labeled by **MPP** (Fig. 5C).

### sCIP-TMT identifies known and novel ligandable cysteines
Comparison to our previous dataset generated using **KB02** MS1-based quantification[15,16,19,105] revealed substantial overlap between the cysteines identified by both approaches (Fig. S12) together with generally good concordance ($r^2 = 0.63$) in the measured ratios, with some unavoidable ratio compression observed for the sCIP-TMT dataset (Fig. 5D), which was acquired using FAIMS-MS2. Further supporting the fidelity of the sCIP-TMT platform, we observe a similarly strong concordance between our sCIP-TMT **KB02** ligandability ratios and those reported by prior studies[15,16,19,105], as aggregated in the human cysteine database (CysDB)[104] similarly revealed consistent ratios (Fig. S13).

Exemplifying established labeling sites, we observe that Cys32 in GSTO1 was labeled to near completion by both **KB02** and **KB10**, consistent with the high ligandability of this cysteine, as reported by a number of previous studies[15,16,19,106](Fig. 5B). Additional targets that proved highly consistent with prior reports include creatine kinase Cys283[107], which is an established target of **KB02** and related analogs, and PIN1 Cys113, for which several highly potent inhibitors have been reported[108,109].

We also compared our datasets generated with MPP to those previously generated using isotopic tandem orthogonal activity-based protein profiling (isoTOP-ABPP)[82] (Fig. 5E). This comparison revealed generally high concordance for cysteines detected by both studies, albeit with some ratio compression for the sCIP-TMT samples (Fig. 5E). Cys249 of the proto-oncogene and adaptor protein CRKL[110] stood out as highly ligandable across all four datasets analyzed, including our previously published MS1-based data and our newly generated sCIP-TMT data (Fig. 5E). To test whether this cysteine was indeed a bona fide compound labeling site, we subjected recombinant CRKL to gel-based activity-based protein profiling (ABPP), using iodoacetamide rhodamine (IA-Rho) to visualize compound- and mutation-induced changes to cysteine labeling. We find that a single point mutation at the cysteine (C249A) completely blocks the labeling of CRKL by IA-Rho and that a similar complete decrease in the signal can be detected for samples subjected to pretreatment with **KB02** or **MPP** (Figs. 5F and S14). These findings corroborate our sCIP-TMT data and additionally reveal a new potential druggable site in a high-value tumor-relevant target. Taken together these findings provide compelling evidence that sCIP-TMT faithfully captures cysteine ligandability sites.

Looking beyond established covalent modification sites, we also asked whether our platform could capture previously unreported ligandable cysteines. Strikingly, nearly all the liganded cysteines (760/789) had been previously identified by CysDB (Fig. S15). Despite this high degree of dataset

overlap, 29 cysteines were uniquely identified as liganded with sCIP-TMT. Exemplary previously unreported liganded sites include Meiosis specific with coiled-coil domain (MEIOC) Cys342 and Akirin-2 Cys3, with the latter located proximal to the 20S proteasome binding motif[11].

Analysis of Akirin-2 and MEIOC spectra with PDV show the presence of the desulfurization characteristic ion, consistent with their identity as cysteine-containing PSMs (Fig. S16). Providing additional confidence in the utility of the desulfurization ion to delineate cysteine-modified peptide spectra, several spectra that lacked this ion were assessed as lower confidence matches (Fig. S17). These findings illustrate the continued opportunities for expanding coverage of the cysteinome, although comparatively modest gains are expected from the continued re-sampling of similar cell line models.

## sCIP-TMT is compatible with N terminal peptide labeling

Motivated by the compelling performance of sCIP-TMT for cysteine chemoproteomics, we opted to also test the broader utility of the approach for applications beyond cysteine labeling, with particular interest in use cases not readily amenable to established isobaric labeling reagents. Pyridine carboxaldehyde (PCA) reagents have recently emerged as a class of useful probes for chemoproteomic detection of protein and peptide N termini[112,113]. PCA-labeling of peptides caps the N terminus with an imidazolidinone moiety, thus blocking peptide N termini, and leaving digested peptides that lack lysine residues inaccessible to derivatization with TMT and related reagents. In contrast, enabled by the clickable handle, we expected that sCIP-TMT should prove compatible with PCA labeling, using the commercially available 5-ethynyl-2-pyridine carboxaldehyde (ethynyl-2PCA), which was recently demonstrated to be compatible with N-terminal proteomics[113]. To test this unique capability of sCIP-TMT, we prepared samples labeled with ethynyl-2PCA, using pre-digested lysates as a model system. After labeling and click conjugation to one of six different sCIP-TMT reagents and enrichment, we subjected samples combined in 1:5:10:15:10:5 ratios to LC-MS/MS analysis. In aggregate, we identified over 700 unique peptide N termini (Fig. S18A) with the reporters observed in their expected ratios (Fig. S18B), demonstrating the capacity of sCIP-TMT for N-terminal proteomic applications.

## Discussion

Here we report the sCIP-TMT platform, which enables high throughput and high coverage cysteine chemoproteomics. To build sCIP-TMT, we first synthesized a customized sCIP–Gly–NH₂ reagent via SPPS that contains biotin, a chemically cleavable DADPS linkage, azide group, and, most importantly, a free amine at the N terminus of the reagent. The presence of this latter moiety allows for straightforward conjugation with commercially available isobaric labeling reagents. Enabled by sCIP–Gly–NH₂, we obtained and deployed a 10-plex set of sCIP-TMT[10] reagents for cysteine chemoproteomics, identifying nearly 20,000 total cysteine residues. We find that the sCIP-TMT[10] platform is compatible with fragment electrophile screening, as demonstrated by our rich datasets of cysteines liganded by the widely utilized scout fragment KB02[15,16,19,106] and electrophilic fragments with more tempered proteome-wide reactivities. Notably, our cysteine chemoproteomic studies reveal that the cysteines labeled by the thio-Michael acceptor MPP fragment show substantial overlap with those labeled by chloroacetamide fragments KB02 and KB10 (Fig. S11), which provides evidence in support of this chemotype as uniquely suited to bridging the chloroacetamide-acrylamide divide. The high accuracy of sCIP-TMT is illustrated by the robust identification of established ligandable cysteine residues, as illustrated by good concordance with our prior studies and those reported in CysDB[16,75,104] (Figs. 5D, E, and S13). Our identification of the desulfurization ion that is present in >97% of sCIP-TMT-modified peptide spectra (Fig. S4) provides additional evidence in corroboration with the prior report[84] that this species has favorable properties to serve as a characteristic ion for modified cysteine peptides. Illustrating the opportunities for future use of this ion in differentiating false positive peptide identifications from novel species, we present high and low-

confidence spectra that feature and lack the desulfurization ion (Figs. S16 and S17). Due to the prevalence of this ion in modified spectra, we envision future work could combine the recently described single-sequence identification (SSI) principles[114] with the desulfurization ion to delineate false positive identifications.

sCIP-TMT offers several important advantages when compared to prior chemoproteomic platforms. The sCIP-TMT workflow's key feature is the early sample pooling, which occurs immediately after click conjugation. While such protein-level sample pooling is common in chemoproteomic platforms that rely on MS1-based quantification[7,13,41,115], nearly all isobaric-reagent-based platforms[15,19,42], with the exception of the aforementioned azido-TMT and anti-TMT approaches, samples are combined after sequence-specific proteolysis. Thus sCIP-TMT substantially streamlines sample preparation compared to these prior methods, as demonstrated by both reduced number of containers and reduced hours of active sample preparation time (Figs. 1 and 4B). We additionally demonstrated that sCIP-TMT also reduced sample-to-sample variance (Fig. 4E), consistent with the previously reported azidoTMT platform[76]. We expect that some of the increased variance in the IA-DTB-TMT-labeled samples could stem from the presence of peptides in the streptavidin-enriched samples, which lack the desthiobiotin modification but feature TMT modifications (Fig. 4F)—these peptides, which carried forward due to non-specific binding to the streptavidin resin are not present in the sCIP-TMT samples (Fig. 4F). Distinct from the azido-TMT and related iodo-TMT[116,117] workflows that require anti-TMT antibody for enrichment, sCIP-TMT is compatible with established avidin-based enrichment platforms. As demonstrated by the recent comparison of chemically cleavable linkers[118], the DADPS moiety used in the sCIP-TMT reagents stands out for its high proteome coverage and compatibility with mild acid elution. Additionally, the active labeling reagent IAA (220 Å2) is smaller than the previously reported desthiobiotin probes, IA-DTB[19] (562 Å2) and DBIA[15,92] (392 Å2), commonly used for these platforms which may prove useful in capturing more sterically shielded cysteine targets. The off-the-shelf compatibility of the sCIP-TMT approach with established data acquisition and analysis pipelines used for TMT and related isobaric labeling strategies obviates requirements for customized software, such as those required for our prior generation of sCIP reagents[75]. Enabled by these many useful features, we expect widespread utility for sCIP-TMT.

Some limitations do remain to be addressed by future enhanced sCIP-TMT platforms. Our sCIP–Gly–NH₂ reagent was obtained in a modest 53% yield, which we expect could be improved by a more stringent assessment of loading onto the 2-chlorotrityl resin, as has been reported previously[119].

An additional potential limitation we do recognize with our current approach is its likely poor amenability to established automation workflows, due to early early sample pooling. We foresee that this challenge could be easily overcome by following a workflow similar to that reported for the recent 96-well plate-based TMT experiments[93,117], as schematized in Fig. S19.

Looking beyond our current study, we envision several immediate use cases for sCIP-TMT. First, while our study used TMT10, expanding to 18-plex multiplexing, by conjugating our sCIP–Gly–NH₂ reagent with TMTpro, should easily enhance the multiplexing capabilities of the sCIP platform. As illustrated by the recently reported chemoproteomic hyperplexing platform[74], we also expect that the incorporation of stable isotopes into our sCIP–Gly–NH₂ reagent through SPPS would allow for the preparation of multiple isobaric sets and similarly efficient hyperplexing. Such hyperplexing strategies will undoubtedly benefit from off-line fractionation to achieve ultra-deep coverage of the proteome, as has been reported for the SLC-ABPP and TMTpro-based platforms[15]. As illustrated by our analysis of ethynyl-2PCA-labeled samples (Fig. S18), we anticipate further unique applications in N-terminal proteomics together with compatibility with other nucleophilic and electrophilic residues for which alkyne-containing probes are available, including lysine[17,30–32], tyrosine[31,33,34], methionine[35,36], histidine[120,121], tryptophan[122], aspartate and glutamate[37,123], phosphoaspartate[124,125], and promiscuously-reactive probes[26,126].

## Methods

### Synthetic procedures

For the preparation of DADPS reagents and sCIP–Gly–NH$_2$ see section (C) of supplementary methods.

### Preparation of sCIP-TMT capture reagents for CuAAC

Each TMT channel (29 mM in MeCN) was mixed in an equimolar ratio with sCIP–Gly–NH$_2$ (29 mM in DMSO) in a 1.5 mL centrifugal tube and let react for 1 h at ambient temperature. After 1 h, 0.5 equivalents of hydroxylamine (10 mM in DMSO) were added and let react for 15 min after which the sCIP-TMT conjugate was ready to be used for CuAAC.

Note: if small amounts of volume are stuck on the side of tubes it is best to briefly centrifuge the sample to allow for complete mixing.

### Proteomic sample preparation for sCIP-TMT profiling of cysteines

HEK293T proteome (100 µL of 2 mg/mL) in a 1.5 mL centrifugal tube was first labeled with IAA (1 µL of 50 mM stock solution in DMSO, final concentration = 500 µM) for 1 h at ambient temperature. CuAAC was performed with the pre-formed sCIP-TMT capture reagent (final concentration = 1 mM), TCEP (2 µL of fresh 50 mM stock in water, final concentration = 1 mM), TBTA (6 µL of 1.7 mM stock in DMSO/t-butanol 1:4, final concentration = 100 µM), CuSO4 (2 µL of 50 mM stock in water, final concentration = 1 mM), and 0.2% SDS for 1 h at ambient temperature. After CuAAC labeling, each sample was treated with 0.5 µL benzonase (Fisher Scientific, 70-664-3) for 30 min at 37 °C. For each 100 µL sample (1 mg/mL protein concentration), 20 µL Sera-Mag SpeedBeads Carboxyl Magnetic Beads, hydrophobic (GE Healthcare, 65152105050250) and 20 µL Sera-Mag SpeedBeads Carboxyl Magnetic Beads, hydrophilic (GE Healthcare, 45152105050250) were mixed and washed with water for three times. The bead slurries were then transferred to the CuAAC samples, and incubated for 5 min at RT with shaking (1000 rpm). Absolute ethanol (400 µL) was added to each sample, and the samples were incubated for 5 min at RT with shaking (1000 rpm). Samples were then placed on a magnetic rack and washed three times with 80% ethanol in water (400 µL). After washing, beads were resuspended in 200 µL, 2 M urea in 0.5% SDS/PBS. DTT (10 µL of 200 mM stock in water, final concentration = 10 mM) was added to each sample and the sample was incubated at 65 °C for 15 min. Then, iodoacetamide (10 µL of 400 mM stock in water, final concentration = 20 mM) was added and the solution was incubated for 30 min at 37 °C with shaking in the dark. Absolute ethanol (400 µL) was added to each sample, and the samples were incubated for a further 5 min at RT with shaking (1000 rpm). Beads were washed three times with 80% ethanol in water (400 µL). Next, beads were resuspended in 200 µL 2 M urea in PBS and 2 µL trypsin solution (Worthington Biochemical, LS003740, 1 mg/mL in 666 µL of 50 mM acetic acid and 334 µL of 100 mM CaCl2) was added. Digest was overnight at 37 °C with shaking. After digestion, ~4 mL acetonitrile (>95% of the final volume) was added to each sample, and the mixtures were incubated for 10 min at RT with shaking (1000 rpm). The beads were then washed three times with 1 mL acetonitrile each with a magnetic rack. Peptides were eluted from SP3 beads with 50 µL of 2% DMSO in Molecular Biology Grade (MB) water for 30 min at 37 °C with shaking (1000 rpm). The elution was repeated with 50 µL of 2% DMSO in MB water. Two eluants were combined. Samples were then enriched as described below.

### Streptavidin enrichment of sCIP-TMT-labeled peptides

For each 200 µL sample of 2 mg/mL cellular lysates, 50 µL of Streptavidin Agarose resin slurry (Pierce, 20353) was washed one time in 8 mL PBS and then resuspended in 500 µL PBS. Peptide solutions eluted from SP3 beads were then transferred to the Streptavidin Agarose resin suspension, and the samples were rotated for 2 h at RT. After incubation, the beads were pelleted by centrifugation (5,000 g, 1 min) and washed twice with 1 mL PBS each and then twice with 1 mL water each. Bound peptides were eluted via acidic cleavage of the DADPS linkage using 200 µL of 2% formic acid in MB water for 30 min at ambient temperature. The elution was repeated once more

with 80% acetonitrile in MB water for 2 min at ambient temperature. The combined eluants were dried (SpeedVac), then reconstituted with 5% acetonitrile and 1% FA in MB water, and analyzed by LC-MS/MS.

### Biological and sample preparation

For additional biological and sample preparation details such as IA-DTB labeling, TMT-labeling, electrophilic small-molecule compound treatment, protein expression and purification, and gel-based ABPP analysis see section (D) biology methods in the supplementary methods.

### Liquid-chromatography tandem mass spectrometry (LC-MS/MS) acquisition

Peptide samples were analyzed by LC-MS/MS using a Thermo Scientific™ Orbitrap Eclipse™ Tribrid™ mass spectrometer or coupled with a High Field Asymmetric Waveform Ion Mobility Spectrometry (FAIMS) Interface via injection of 400–800 ng peptide per sample. Peptides were fractionated S21 online using an 18 cm long, 100 µM inner diameter (ID) fused silica capillary packed in-house with bulk C18 reversed phase resin (particle size, 1.9 µm; pore size, 100 Å; Dr. Maisch GmbH). The 70 and 180-minute water-acetonitrile gradient was delivered using a Thermo Scientific™ EASY-nLC™ 1200 system at different flow rates (buffer A: water with 3% DMSO and 0.1% formic acid and buffer B: 80% acetonitrile with 3% DMSO and 0.1% formic acid). The detailed 70-minute gradient includes 0–5 min from 3% to 10% at 300 nL/min, 5–64 min from 10% to 50% at 220 nL/min, and 64–70 min from 50% to 95% at 250 nL/min buffer B in buffer A. The detailed 180-minute gradient includes 0–5 min from 2% to 6% at 300 nL/min, 5–151 min from 6% to 50% at 220 nL/min, and 151–180 min from 50% to 95% at 250 nL/min buffer B in buffer A. Data was collected with charge exclusion (1, 8, >8). Data was acquired using a data-dependent acquisition (DDA) method consisting of a full MS1 scan (resolution = 120,000) followed by sequential MS2 scans (resolution varied by experiment) to utilize the remainder of the 1-s cycle time. Precursor isolation window and NCE were set as described in the study. Conditions of Liquid-chromatography (LC) Parameter Condition Column 100 µM ID fused silica capillary packed in-house with bulk C18 reversed phase resin (particle size, 1.9 µm; pore size, 100 Å; Dr. Maisch GmbH). For additional details on mass spectrometry methods see section (E) and Table S2 in the supplementary methods.

### Protein and peptide identification

Raw data collected by LC-MS/MS were searched with MSFragger (v3.7 and v3.8) and FragPipe (v19.0-19.2 and v20.0). For closed search, the "default" proteomic workflow was loaded in FragPipe and these default values were used for all settings, except as noted below. Precursor and fragment mass tolerance was set as 20 ppm. Missed cleavages were allowed up to 1. A human protein database was downloaded from UniProtKB on [January 1st, 2020] using FragPipe, containing reviewed sequences and common contaminants, with 37110 total entries. Digestion was performed in MSFragger using the 'strict trypsin' (i.e., allowing cleavage before P) setting, peptide length was set 7–50, and peptide mass range was set to 500–5000. Cysteine residues were searched with differential modifications as described in the study, allowing a max of 2 per peptide. Cys carbamidomethylation was additionally set as a variable modification (max 2 per peptide). For the labile search, a single modification mass was set as a mass offset and was restricted to cysteines. Labile search mode was enabled with Y ion masses and diagnostic fragment masses set as in Figs. 2 and S4 for different proteomic samples, and diagnostic ion minimum intensity of 0.02. PTM-Shepherd was enabled for fragment analysis. PSM validation, protein inference, and FDR filtering were performed in PeptideProphet, ProteinProphet, and Philosopher, respectively, in FragPipe using default settings. Results were generated using the TMT integrator output where the best PSM was set to false, allow unlabeled was set to true, and the mod tag was set to the mass of the intact cysteine modification C(638.40614) for sCIP-TMT. For IA-DTB and TMT-labeled samples a variable cysteine modification in MSFragger was set at C(455.2743) for IA-DTB and fixed modifications were set at K(229.1629) and N-Term peptide(229.1629) for TMT. In the TMT integrator, the

samples were searched as above with the mod tag set to the mass of the IA-DTB modification C(455.2743). For ethynyl-2PCA-labeled samples a fixed modification in MSFragger was set at N-term peptide(614.3492) and the mod tag in the TMT integrator was set to the mass of the intact modification N-term(614.3492).

## Data availability

Mass spectrometry data files are available in Supplementary Data 1–4 and in the PRIDE repository: PXD049154 (See Table S3 for File details).

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

## Acknowledgements

This study was supported by a 2021 Noble Family Innovation Fund Seed Project Award (K.M.B.), DOD-Advanced Research Projects Agency (DARPA) 1DP2GM146246-01 (K.M.B.), National Institutes of Health DP2 OD030950-01 (K.M.B.), National Institute of General Medical Sciences T32 GM067555-11 (N.R.B.). We thank all members of the Backus lab for helpful suggestions, as well as the UCLA Proteome Research Center for assistance with mass spectrometry-based proteomic data collection.

## Author contributions

N.R.B. and K.M.B. conceived of the project. N.R.B. performed all experiments and collected and processed all the data. N.R.B. and K.M.B. wrote the manuscript.

## Competing interests

K.M.B. is a Guest Editor for *Communications Chemistry* but was not involved in the editorial review of, or the decision to publish this article. The authors declare no competing interests.
