## [Peer Review File · Communications Chemistry]

Reviewers' comments:

Reviewer #1 (Remarks to the Author):

The manuscript by Nikolas Burton and Keriann Backus describes a novel trifunctional linker for chemical proteomics with focus on profiling of reactive cysteines to explore covalent cysteine reactive ligands. The key features of the linker are silane-based cleavable group and a TMT-tag. The silane-based cleavable moiety was previously described, now combination with TMT-tags is novel and might find a good usage within the chemical proteomics community. The application of the sCIP-TMT linker is shown to shorten the MS-sample preparation workflow by early combination of replicates/conditioned samples. Overall, authors present convincing validation of the sCIP-TMT linker functionality and some application scope. The reviewer appreciates the detail MS spectra analysis and diagnostic ion mining as a great way to elucidate the fragmentation process. Although the manuscript is well structured with a clear sections describing the workflow and application, there are some shortcomings and it seems that it is narrowly focused on covalent probe/chemical proteomics readers with somewhat restricted gain of workflow throughput. I would suggest to extend the scope of the workflow to ensure it is compatible with standard automatised procedures, scale down and to identify the method limitations.

Major comments:

- 1) The main principal advantage of sCIP-TMT over the 'late' introduction of TMT-tags, which does not necessitate for synthesis of specialised linker, would be less sample-to-sample variance (Figure 1A). The authors should compare the variance in a side-by-side experiment starting from a single lysate incubated either with IAA and reacted with sCIP-TMT or with desthiobiotin iodoacetamide (DBIA) probe and after enrichment with standard TMT-tags.
- 2) To identify the limitations of the method, it would be interesting to test the workflow with decreasing amount of starting total protein. Does the coverage/total number of identified reactive cysteines increases with increasing amount. Is that superior to previously published methods?
- 3) By introduction of the affinity-tag and TMT-tag in the single reaction might improve overall enrichment-labelling efficiency and thus total amount of enriched and labelled peptides. Does it lead as well to a possibility to shorten the LC-MS/MS gradients? What is the coverage using the 60 and 30 min gradients?
- 4) The cysteines are highly reactive and can be labelled with high selectivity and yield, which works well together with sCIP-TMT. It is anticipated that the sCIP-TMT may be applied to other amino acid side chains, however, experimental comparison with any other reactive probe would

solidify this notion.

Minor comments:

- 1) I would suggest a more descriptive title by replacing the sCIP abbreviation.
- 3) Line 30, give the number of validated compounds, the set of compounds might be more appropriate than library.
- 2) DADPS abbreviation should be explained when it is first mentioned on line 90.
- 3) In Figure 1, the structures of the IA-DTB and IAA should be included. IAA is commonly used for iodoacetamide as well.
- 4) Figure 1, depicts the workflow to be performed in single tubes. Can it be done in well-plate format? Was it done in well-plate format, it is not completely clear from the manuscript and SI.
- 5) On line 28, the TMT needs to be introduced in full name.
- 6) Line 156, concentration and incubation time of IAA should be mentioned.
- 7) Line 158, was the FAIMS used in LC-MS/MS analysis?
- 8) Line 172, is there a quantitative measure on how many spectra the TMT ion was dominant?
- 9) From line 285, I would suggest to unify the nomenclature of proteins of interest to use e.g. Uniprot gene names. With the full name when it appears for the first time in the text (GSTO1 line 282 x line 312)
- 10) The SI need a proofreading to correct the chemical formulas' subscripts.

Reviewer #2 (Remarks to the Author):

This manuscript describes the development of sCIP-TMT capture reagents and platform for covalent chemoproteomic target profiling. By combining the sCIP reagent and amine-based labeling strategy, the authors show that sCIP-TMT is capable of conjugating TMT to alkyne labeled proteins, which allows for early-stage isobaric labeling. This method has great application potential for cysteine-reactive electrophilic fragment screening.

The overall technical quality in this manuscript is high. Introducing isobaric tags onto labeling sites by click chemistry for early-stage sample pooling is novel. The sCLIP-TMT workflow and its related reagents will be of utility to researchers who are interested in cysteine or other amino acids-centric studies, and I therefore recommend publication after addressing the following minor concerns.

1. The authors could further compare sCIP with previous datasets collected from MS1-based or DIA-based quantification for better understanding the impact of ratio compression on the final datasets.

2. The authors should attempt to validate several liganded cysteines (presented in Figure 4D) using gel-based approaches.

Reviewer #3 (Remarks to the Author):

The authors establish a novel Cysteine reactive compound that is fully formed (mass spectrometry distinguishable mass labels and enrichment handle) early in the process of labeling the reactive cysteines. Prior developments used click chemistry to incorporate the TMT-like labels at a later timepoint and required separate enrichment per sample. Especially the latter is problematic as the labelled fraction generally speaking is of low amounts and it is easy to lose interesting material this way. The enrichment handle additionally is well designed, as elution of biotin from streptavidin is in our laboratory a recipe for failure and likewise its presence on the eventual peptides will contaminate the fragmentation spectra to such a degree that identification will become difficult. The incorporated cleavage site efficiently allows removal of the biotin to elute the bound peptide fraction.

The authors 'sell' the advance mostly in terms of speed up of the experimental workflow, which is a valid point, but I would like to point out that the reagent itself also represents a very nice leap from previous efforts. The manuscript is very well written and the advance it represents is more than convincing for publication in Communications Chemistry. I whole-heartedly support its acceptance after fixing the following small remarks:

1. It would be insightful to know the size of the reagent and how that compares to the already established reagents (e.g. such a measure can be obtained with 'get_area' in pyMol). This can be seen as a measure of the to be expected steric hindrance.
2. 53% yield during the synthesis step seems low and would definitely need to be improved to enable large scale production. The authors should add a couple of sentences about how they would go about increasing this.
3. Supplementary material: (1) The actual NMR spectra were missing for the intermediate and final compounds, these should be added (annotated) to the supplementary material; (2) the HPLC trace of the conversion to sCIP-TMT is in principle convincing, but would get further credence with either an NMR and/or accurate intact mass; (3) the HRMS figure on page 13 needs to contain both the chromatographic trace as well as the m/z spectrum to show the intact mass.
4. It is unclear to me whether short (how long is this ?) incubation with 0.5 equivalent of hydroxylamine sufficiently quenches the TMT reagent. Can the authors comment on experiments they did to show this? Additionally, would it not make sense to remove the excess TMT labels through chromatography to reduce contaminants in the final biological sample?
5. Desulfurization of labeled cysteines occurring in >97% of the cases seems to me a very attractive approach to remove all false positives. FDR approaches in proteomics remain an

estimate and even cutting at 1% for large datasets in actual fact could be 2.5% for most search engines. Can the authors separately look at the identifications missing the desulfurization and get a feel of whether or not these are really false positives?

6. Pg 12, line 12, "Importantly, the measured reporter ion intensity ratios were observed to closely match the expected values." <- this is missing a reference to the figure 3c.

7. Pg 14: "revealed substantial overlap between the cysteines identified by both approaches (Figure S10) together with high concordance ($r^2 = 0.63$) in the measured ratios". <- this language needs to be toned down. Roughly 50% overlap according to Fig S10 is not substantial and a r^2 of 0.63 is not high concordance (in this reviewer's opinion such a statement would need $r^2=1$ as it's easy to make mistakes). Just report the quantities and let the reader make up their mind.

8. Page 15: 29 novel cysteines, would require a little bit more in-depth analysis. For example, do they contain the desulfurization diagnostic ion; and an overview in the supplementary material. This is new information after all.

Reviewers' comments:

Reviewer #1 (Remarks to the Author):

The manuscript by Nikolas Burton and Keriann Backus describes a novel trifunctional linker for chemical proteomics with focus on profiling of reactive cysteines to explore covalent cysteine reactive ligands. The key features of the linker are silane-based cleavable group and a TMT-tag. The silane-based cleavable moiety was previously described, now combination with TMT-tags is novel and might find a good usage within the chemical proteomics community. The application of the sCIP-TMT linker is shown to shorten the MS-sample preparation workflow by early combination of replicates/conditioned samples. Overall, authors present convincing validation of the sCIP-TMT linker functionality and some application scope. The reviewer appreciates the detail MS spectra analysis and diagnostic ion mining as a great way to elucidate the fragmentation process. Although the manuscript is well structured with a clear sections describing the workflow and application, there are some shortcomings and it seems that it is narrowly focused on covalent probe/chemical proteomics readers with somewhat restricted gain of workflow throughput. I would suggest to extend the scope of the workflow to ensure it is compatible with standard automatisations procedures, scale down and to identify the method limitations.

We thank this reviewer for their positive and constructive assessment of our manuscript. In our point-by-point response, below, we have incorporated their feedback, which we believe substantially strengthens our manuscript.

Major comments:

1) The main principal advantage of sCIP-TMT over the 'late' introduction of TMT-tags, which does not necessitate for synthesis of specialised linker, would be less sample-to-sample variance (Figure 1A). The authors should compare the variance in a side-by-side experiment starting from a single lysate incubated either with IAA and reacted with sCIP-TMT or with desthiobiotin iodoacetamide (DBIA) probe and after enrichment with standard TMT-tags.

As suggested by this reviewer, we have added a direct-head-to-head comparison between the sCIP-TMT and DBIA+TMT approaches, and the results from this experiment are now included in **Figure 3** and **Figure S8**. These experiments revealed comparable enrichment efficiency for each method together with decreased variance for the sCIP-TMT approach, which is consistent with the findings reported for the recently reported azidoTMT platform (J. Proteome Res. 2023, 22, 2218–2231).

Figure 3. sCIP-TMT faithfully quantifies cysteine ratios with decreased sample preparation times. (A) Peptide, cysteine, and protein coverage of sCIP-TMT¹⁰ labeled samples mixed 1:1 analyzed using FAIMS-MS². (B) Analysis of time (hours) and tubes saved using the sCIP-TMT workflow as multiplex channels increases. (C) Comparison of ratios for samples mixed in both 1:1 and 1:5:10:15 ratios analyzed using FAIMS-MS². (D) Comparison of ratios for samples mixed in both 1:1 and 1:5:10:15 ratios analyzed using SPS-MS³. (E) Comparison of sCIP-TMT and IA-DTB/TMT coefficient of variance for samples mixed in 1:1 or 1:4 ratios. (F) Percentage of enriched cysteine peptides which contain either a sCIP-TMT modification (green), a TMT modification (purple), or a IA-DTB modification (striped purple). Box plots display 5th percentile, first quartile (Q1), median,

third quartile (Q3), and 95th percentile values of the sample. For panels C-E, n=3 biological replicates. All MS data can be found in **Table S3**.

Figure S8. sCIP-TMT126 and sCIP-TMT127N labeled samples were combined either 1:1 or 1:4 directly after click, prior to sample preparation. IA-DTB labeled samples were taken through the sample preparation separately, enriched with streptavidin, then labeled with either TMT126 or TMT127N before being combined in 1:1 and 1:4 ratios. (A) Peptides, cysteines, and protein counts for samples analyzed using either sCIP-TMT analysis or TMT sample preparation. (B) Comparison of ratios for samples mixed in both 1:1 and 1:4 ratios using sCIP-TMT and TMT sample preparation. Box plots display 5th percentile, first quartile (Q1), median, third quartile (Q3), and 95th percentile values of the sample. For sCIP-TMT analysis n=5 biological replicates, and for TMT analysis n=4 biological replicates. All MS data can be found in **Table S4**.

2) To identify the limitations of the method, it would be interesting to test the workflow with decreasing amount of starting total protein. Does the coverage/total number of identified reactive cysteines increases with increasing amount. Is that superior to previously published methods?

We thank this reviewer for pointing out a need for addressing a key possible strength (or limitation) of our method, namely the amount of input required to achieve high coverage. As was recently reported by Gygi and coworkers, low input amounts are now feasible for chemoproteomics, using the SLC-TMT method (<https://doi.org/10.1016/j.chembiol.2023.11.015>). In our head-to-head comparison with TMT, we observe that sCIP-TMT does afford a modest increase in coverage (**Figure S8**), supporting increased performance. No difference in enrichment efficiency was however observed for this comparison, which indicates that there likely will be no major difference in performance between the two workflows. As scalability is a major consideration for automatization, we have added the following text to the discussion, “One potential limitation we do recognize with our current approach is its likely poor amenability to established automation workflows, due to early the early sample pooling. We foresee that this challenge could be easily overcome through following a workflow similar to that reported for the recent 96-well plate-based TMT experiments^{93,117}, as schematized **Figure S18**.”

Figure S18. Comparison of (A) TMT sample prep workflow and (B) hypothesized sCIP-TMT sample preparation workflow in a 96-well format to allow for automation-compatibility.

3) By introduction of the affinity-tag and TMT-tag in the single reaction might improve overall enrichment-labelling efficiency and thus total amount of enriched and labelled peptides. Does it lead as well to a possibility to shorten the LC-MS/MS gradients? What is the coverage using the 60 and 30 min gradients?

We concur with this reviewer that the opportunities for improving coverage with sCIP-TMT are indeed exciting. Disappointingly, we do not, however, observe increased enrichment efficiency when compared to IA-DTB+TMT, as shown in **Figure 3F** and thus we do not expect major increases in coverage that would warrant shorter gradients for acquisition. We have modified the text to the following.” Supporting no major performance differences between the two workflows, similar capture efficiency was observed for both workflows, with >80% of all identified peptides harboring a biotin/desthiobiotin modification (**Figure 3F**).”

Additionally, given how high performing the IA-DTB+TMT platform has proven to be, we do not expect that sCIP-TMT will substantially outperform this excellent established workflow, although our head-to-head comparison is suggestive of modest gains in coverage and decreased variance. That being said, we expect that the major advantage of the sCIP-TMT approach is to access applications that are not readily accessible to TMT. To illustrate these unique opportunities, we have also deployed sCIP-TMT on n-terminal proteomics, as shown in **Figure S16**, which is a use-case that is inaccessible to TMT, due to the absence of TMT labeling sites in many peptides derived from protein n-termini.

Figure S16. (A) Peptides, n-termini, and protein counts for samples analyzed using ethynyl-2PCA labeling and sCIP-TMT sample preparation. (B) Ratios for samples labeled with the indicated sCIP-TMT reagent and mixed in a 1:5:10:15:10:5 ratio. Box plots display 5th percentile, first quartile (Q1), median, third quartile (Q3), and 95th percentile values of the sample. Error bars on bar plots display standard deviation. n=2 biological replicates. All MS data can be found in **Table S3**.

4) The cysteines are highly reactive and can be labelled with high selectivity and yield, which works well together with sCIP-TMT. It is anticipated that the sCIP-TMT may be applied to other amino acid side chains, however, experimental comparison with any other reactive probe would solidify this notion.

We thank the reviewer for this excellent suggestion, which prompted us to identify use-cases that are incompatible with TMT and other established isobaric labeling approaches. Illustrating the the generalizability and unique applications of the sCIP-TMT platform, we generated proof-of-concept n-terminal proteomics datasets using the recently published ethynyl-2PCA probe (<https://doi.org/10.1016/j.chembiol.2023.09.009>). In exciting results, we find that sCIP-TMT is indeed compatible with EPCA-labeled peptide capture, which we expect will open up new opportunities for future n-terminal proteomics applications. Results of this analysis are shown in **Figure S16** and we have added the following text to the manuscript: *“Motivated by the compelling performance of sCIP-TMT for cysteine chemoproteomics, we opted to also test the broader utility of the approach for applications beyond cysteine labeling, with particular interest in use cases not readily amenable to established isobaric labeling reagents. Pyridine carboxaldehyde (PCA) reagents have recently emerged as a class of useful probes for chemoproteomic detection of protein and peptide n-termini^{112,113}. PCA-labeling of peptides caps the n-terminus with an imidazolidinone moiety, thus blocking peptide n-termini, and leaving digested peptides that lack lysine residues inaccessible to derivatization with TMT and related reagents. In contrast, enabled by the clickable handle, we expected that sCIP-TMT should prove compatible with PCA labeling, using the commercially available 5-ethynyl-2-pyridine carboxaldehyde (ethynyl-2PCA), which was recently demonstrated to be compatible with n-terminal proteomics¹¹³. To test this unique capability of sCIP-TMT, we prepared samples labeled with ethynyl-2PCA, using pre-digested lysates as a model system. After labeling and click conjugation to one of six different sCIP-TMT reagents and enrichment, we subjected samples combined in 1:5:10:15:10:5 ratios to LC-MS/MS analysis. In aggregated, we identified over 700 unique peptide n-termini (**Figure S16A**) with the reporters observed in their expected ratios (**Figure S16B**), demonstrating the capacity of sCIP-TMT for n-terminal proteomic applications.”*

Minor comments:

- 1) I would suggest a more descriptive title by replacing the sCIP abbreviation. Our intention for the title was to capture the essence of the manuscript for a broad audience without being overly technical. As such, we would prefer to leave the title unchanged, but will defer to this reviewer and the editorial staff if there is a strong preference for a different title.
- 2) Line 30, give the number of validated compounds, the set of compounds might be more appropriate than library.
We thank the reviewer for bringing this to our attention and have updated this sentence to read *“By screening a focused set of four cysteine-reactive electrophiles...”*
- 3) DADPS abbreviation should be explained when it is first mentioned on line 90.
We thank the reviewer for bringing this to our attention and we have added the full name to the text such that the sentence now reads, *“...a chemically cleavable dialkoxydiphenylsilane (DADPS) group..”*
- 4) In Figure 1, the structures of the IA-DTB and IAA should be included. IAA is commonly used for iodoacetamide as well.
We thank the reviewer for identifying a point of potential confusion and have added the structure of IA-DTB and DBIA next to the structure of IAA in **Figure S1**.
- 5) Figure 1, depicts the workflow to be performed in single tubes. Can it be done in well-plate format? Was it done in well-plate format, it is not completely clear from the manuscript and SI.
We thank the reviewer for identifying a point of potential confusion. In all our experiments we used 1.5 mL centrifugal tubes, which we have clarified in the methods section. We do not foresee any substantial difficulties transitioning this workflow into a 96-well plate format, given the substantial literature precedent for related studies. As this study focused on the reagent design, we hope that this reviewer will concur that acquiring an automated liquid handler, which we do not yet have in our lab, together with setting up an automated version of this workflow are beyond the scope of the current study. We do agree with this reviewer that such work would be an exciting future application of our reagents and to address these points, we have added the following text to the discussion and additionally added **Figure S19** to show how we expect that sCIP-TMT could be adapted to a plate format for automation: *“One potential limitation we do recognize with our current approach is its likely poor amenability to established automation workflows, due to early the early sample pooling. We foresee that this challenge could be easily overcome through following a workflow similar to that reported for the recent 96-well plate-based TMT experiments^{93,117}, as schematized **Figure S18**.”*
- 6) On line 28, the TMT needs to be introduced in full name.
We thank the reviewer for bringing this to our attention and we have added the full name to the text such that the sentence now reads, *“sCIP-TMT pairs a custom click-compatible sCIP capture reagent that is readily functionalized in high yield with commercially available tandem mass tags (TMT) reagents.”*
- 7) Line 156, concentration and incubation time of IAA should be mentioned.
We thank the reviewer for their suggestion and have added this information to the text which now reads, *“Following the workflow shown in **Figure 1B**, cell lysates were capped with the pan-*

cysteine reactive iodoacetamide alkyne (IAA) probe (500 μ M, 1hr) followed by click conjugation to the preformed sCIP-TMTzero conjugate.”

8) Line 158, was the FAIMS used in LC-MS/MS analysis?

We have updated **Table S6** to contain the method details for all MS experiments.

9) Line 172, is there a quantitative measure on how many spectra the TMT ion was dominant?

Analysis of the diagnostic ion mining data revealed the TMT was the more intense ion compared to the sulfurization ion in >91% of spectra. We have updated the text which now reads, *“Inspection of the mass spectra using FragPipe-PDV^{85,86} confirmed the presence of these fragment ions with the TMT reporter being the dominant ion in nearly all spectra (>91%) (Figure 2C).”*

9) From line 285, I would suggest to unify the nomenclature of proteins of interest to use e.g. Uniprot gene names. With the full name when it appears for the first time in the text (GSTO1 line 282 x line 312)

We thank the reviewer for bringing this to our attention and have added the full name of the gene to the first instance that it appears in the text.

10) The SI need a proofreading to correct the chemical formulas' subscripts.

We have carefully edited the supporting information to ensure that the chemical formulas are correct.

Reviewer #2 (Remarks to the Author):

This manuscript describes the development of sCIP-TMT capture reagents and platform for covalent chemoproteomic target profiling. By combining the sCIP reagent and amine-based labeling strategy, the authors show that sCIP-TMT is capable of conjugating TMT to alkyne labeled proteins, which allows for early-stage isobaric labeling. This method has great application potential for cysteine-reactive electrophilic fragment screening.

The overall technical quality in this manuscript is high. Introducing isobaric tags onto labeling sites by click chemistry for early-stage sample pooling is novel. The sCLIP-TMT workflow and its related reagents will be of utility to researchers who are interested in cysteine or other amino acids-centric studies, and I therefore recommend publication after addressing the following minor concerns.

We thank this reviewer for their strong endorsement of our manuscript. We have incorporated their excellent suggestions into our revised submission, as detailed below.

1. The authors could further compare sCIP with previous datasets collected from MS1-based or DIA-based quantification for better understanding the impact of ratio compression on the final datasets.

To further benchmark our findings against published datasets, we have incorporated the following comparisons. First, we have compared our ligandability dataset generated here using KB02 scout fragment treatments of HEK293T cell lysates and FAIMS-MS2-sCIP-TMT analysis with comparable data that we had previously acquired for KB02 with sample preparation using first generation heavy and light sCIP reagents and MS1-based quantification (<https://doi.org/10.1021/jacs.3c05797>). We have additionally compared our MPP FAIMS-MS2-sCIP-TMT dataset to our recently acquired dataset for MPP that was prepared by isoTOP-ABPP with FAIMS acquisition and MS1 quantification (<https://doi.org/10.1101/2023.10.25.563785>). This

comparison is now shown in **Figure 4E**. Both comparisons show generally high concordance for shared cysteines, albeit with some ratio compression for the sCIP-TMT samples.

- The authors should attempt to validate several liganded cysteines (presented in Figure 4D) using gel-based approaches.

As suggested by this reviewer, to further confirm the validity of our findings, we additionally performed gel-based ABPP on the proto-oncogene and adaptor protein CRKL, which was identified as highly ligandable by our focused screen. We find that a single point mutation at the cysteine (C249A) completely blocks labeling of CRKL by iodoacetamide rhodamine and that a similar complete decrease in signal can be detected for samples subjected to pretreatment with **KB02** or **MPP**, corroborating our sCIP-TMT data. The results of this experiment are displayed in **Figure 4F** and are copied below.

Reviewer #3 (Remarks to the Author):

The authors establish a novel Cysteine reactive compound that is fully formed (mass spectrometry distinguishable mass labels and enrichment handle) early in the process of labeling the reactive cysteines. Prior developments used click chemistry to incorporate the TMT-like labels at a later timepoint and required separate enrichment per sample. Especially the latter is problematic as

the labelled fraction generally speaking is of low amounts and it is easy to lose interesting material this way. The enrichment handle additionally is well designed, as elution of biotin from streptavidin is in our laboratory a recipe for failure and likewise its presence on the eventual peptides will contaminate the fragmentation spectra to such a degree that identification will become difficult. The incorporated cleavage site efficiently allows removal of the biotin to elute the bound peptide fraction.

The authors 'sell' the advance mostly in terms of speed up of the experimental workflow, which is a valid point, but I would like to point out that the reagent itself also represents a very nice leap from previous efforts. The manuscript is very well written and the advance it represents is more than convincing for publication in Communications Chemistry. I whole-heartedly support its acceptance after fixing the following small remarks:

We thank the reviewer for the positive review of our manuscript and the constructive points raised. We have addressed these concerns below:

1) It would be insightful to know the size of the reagent and how that compares to the already established reagents (e.g. such a measure can be obtained with 'get_area' in pyMol). This can be seen as a measure of the to be expected steric hindrance.

We performed this analysis in pymol and have added the following text to the discussion, *"Additionally, the active labeling reagent IAA (220 Å²) is smaller than the desthiobiotin probes, IA-DTB (562 Å²) and DBIA (392 Å²), commonly used for these platforms which may prove useful in capturing more sterically shielded cysteine targets."*

2) 53% yield during the synthesis step seems low and would definitely need to be improved to enable large scale production. The authors should add a couple of sentences about how they would go about increasing this.

We agree with the reviewer that yield can always be a pitfall of production for consumable materials and have added the following text to the discussion to address these points: *"Our sCIP-Gly-NH₂ reagent was obtained in modest 53% yield, which we expect could be improved by more stringent assessment of loading onto 2-chlorotrityl resin, as has been reported previously (<https://doi.org/10.1021/acs.jchemed.3c00186>)."*

3. Supplementary material: (1) The actual NMR spectra were missing for the intermediate and final compounds, these should be added (annotated) to the supplementary material; (2) the HPLC trace of the conversion to sCIP-TMT is in principle convincing, but would get further credence with either an NMR and/or accurate intact mass; (3) the HRMS figure on page 13 needs to contain both the chromatographic trace as well as the m/z spectrum to show the intact mass.

We have added the NMR spectra for the synthesis of the solid-phase compatible DADPS reagent and the full mass spectra for the sCIP-TMTzero conjugate showing the intact mass to the supplementary information.

4. It is unclear to me whether short (how long is this ?) incubation with 0.5 equivalent of hydroxylamine sufficiently quenches the TMT reagent. Can the authors comment on experiments they did to show this? Additionally, would it not make sense to remove the excess TMT labels through chromatography to reduce contaminants in the final biological sample?

We opted to use these conditions as they are what is typically used in TMT experiments where after TMT labeling excess reagent is quenched with 5% hydroxylamine for 15 minutes. Notably, we opted against chromatography as this would add additional steps to the workflow and possibly decrease the recovery of these costly reagents. Furthermore, the excess quenched TMT reagents are removed during the SP3 decontamination, so we do not expect the excess residual reagents to complicate the sample preparation workflow. We have added the following reference to this portion of the text: doi.org/10.1016/j.xpro.2021.100458

5. Desulfurization of labeled cysteines occurring in >97% of the cases seems to me a very attractive approach to remove all false positives. FDR approaches in proteomics remain an estimate and even cutting at 1% for large datasets in actual fact could be 2.5% for most search engines. Can the authors separately look at the identifications missing the desulfurization and get a feel of whether or not these are really false positives?

As this reviewer astutely notes, characteristic ions can prove useful for gaining confidence in identified spectra, as described in response #8 below, and also for possibly delineating false IDs, as noted here. To incorporate these points into our manuscript, we have modified the discussion to include the following: We have added the following text to the discussion to address the possibility of implementing this, *“Our identification of the desulfurization ion that is present in >97% of sCIP-TMT-modified peptide spectra (Figure S4) provides additional evidence in corroboration with the prior report⁹⁴ that this species has favorable properties to serve as a characteristic ion for modified cysteine peptides. Illustrating the opportunities for future use of this ion in differentiating false positive peptide identifications from novel species, we present high and low confidence spectra that feature and lack the desulfurization ion. (Figure S15 and S16). Due to the prevalence of this ion in modified spectra, we envision future work could combine the recently described single-sequence identification (SSI) principles¹¹⁴ with the desulfurization ion to delineate false positive identifications.”*

”

6. Pg 12, line 12, “Importantly, the measured reporter ion intensity ratios were observed to closely match the expected values.” <- this is missing a reference to the figure 3c.

We thank the reviewer for noticing this typo. We have added the reference to Figure 3C such that now the text reads, *“..observed to closely match the expected values (Figure 3C).”*

7. Pg 14: “revealed substantial overlap between the cysteines identified by both approaches (Figure S10) together with high concordance ($r^2 = 0.63$) in the measured ratios”. <- this language needs to be toned down. Roughly 50% overlap according to Fig S10 is not substantial and a r^2 of 0.63 is not high concordance (in this reviewer’s opinion such a statement would need $r^2=1$ as it’s easy to make mistakes). Just report the quantities and let the reader make up their mind.

We agree with this reviewer that it is important to not overstate our findings, as there is some variance in the ratios obtained. To address this point, we have modified the text to read, *“together with generally good concordance ($r^2 = 0.63$) in the measured ratios”*

8. Page 15: 29 novel cysteines, would require a little bit more in-depth analysis. For example, do they contain the desulfurization diagnostic ion; and an overview in the supplementary material. This is new information after all.

We concur with this reviewer that the desulfurization characteristic ion represents an exciting opportunity to further enhance discovery of novel cysteine-peptide containing PSMs. We were delighted to observe that manual inspection of spectra for several of these novel cysteines did indeed contain the desulfurization products, which are now presented in **Figure S15**, with the revised text reading, *“Exemplary novel liganded sites include Meiosis specific with coiled-coil domain (MEIOC) Cys342 and Akirin-2 Cys3, with the latter located proximal to the 20S proteasome binding motif¹¹. Analysis of Akirin-2 and MEIOC spectra with PDV show the presence of the desulfurization characteristic ion, consistent with their identify as cysteine-containing PSMs (Figure S15).”*

Figure S15. Identification of desulfurization ion in Akirin-2 and MEIOC peptide spectra.

REVIEWERS' COMMENTS:

Reviewer #1 (Remarks to the Author):

I would like to thank the authors for addressing the comments. The comparison between sCIP-TMT and IA-DTB/TMT demonstrates the improvement of the protocol. I now fully support the publication of the manuscript.

Reviewer #2 (Remarks to the Author):

The authors have satisfactorily addressed all my concerns. This manuscript is ready for publication.

Reviewer #3 (Remarks to the Author):

No further comments.